# Learning Cooperative Trajectory Representations for Motion Forecasting

**Hongzhi Ruan** [1,2]   **Haibao Yu** [1,3*]   **Wenxian Yang** [1]   **Siqi Fan** [1]   **Zaiqing Nie** [1*]

[1] Institute for AI Industry Research (AIR), Tsinghua University
[2] University of Chinese Academy of Science    [3] The University of Hong Kong
hongzhi.rynn@gmail.com, yuhaibao94@gmail.com, zaiqing@air.tsinghua.edu.cn

## Abstract

Motion forecasting is an essential task for autonomous driving, and utilizing information from infrastructure and other vehicles can enhance forecasting capabilities. Existing research mainly focuses on leveraging single-frame cooperative information to enhance the limited perception capability of the ego vehicle, while under-utilizing the motion and interaction context of traffic participants observed from cooperative devices. In this paper, we propose a forecasting-oriented representation paradigm to utilize motion and interaction features from cooperative information. Specifically, we present V2X-Graph, a representative framework to achieve interpretable and end-to-end trajectory feature fusion for cooperative motion forecasting. V2X-Graph is evaluated on V2X-Seq in vehicle-to-infrastructure (V2I) scenarios. To further evaluate on vehicle-to-everything (V2X) scenario, we construct the first real-world V2X motion forecasting dataset V2X-Traj, which contains multiple autonomous vehicles and infrastructure in every scenario. Experimental results on both V2X-Seq and V2X-Traj show the advantage of our method. We hope both V2X-Graph and V2X-Traj will benefit the further development of cooperative motion forecasting. Find the project at https://github.com/AIR-THU/V2X-Graph.

## 1   Introduction

In recent years, autonomous driving has made significant progress. However, single-vehicle autonomous driving still faces substantial safety challenges due to its limited perception ability. Utilizing external information, such as data from other connected autonomous vehicles and infrastructure sensors through vehicle-to-everything (V2X), has shown great potential to enhance autonomous driving capabilities. In this paper, we focus on motion forecasting, a fundamental task for autonomous driving that has received significant attention in recent years [40, 33, 31, 19]. Specifically, considering currently practical communication conditions, we transmit perception results and input trajectories from the ego vehicle and external sources for cooperative motion forecasting.

Cooperative motion forecasting involves the ego vehicle aggregating its own data with data transmitted from other connected vehicles or infrastructure devices to predict future waypoints for each agent in traffic scenarios. To accommodate limited communication conditions, we consider data in the form of perception results. These perception results form historical trajectories of agents from respective views, termed cooperative trajectories. The autonomous vehicle utilizes these trajectories to enhance its motion forecasting capabilities. High-definition (HD) maps are also used in this task.

To leverage cooperative trajectories for improving motion forecasting performance, two critical issues must be addressed: (1) Observations of the agents from different views may different due to various sensor perspectives and configurations; (2) In the cooperative scenario, there are multi-

---

*Corresponding authors.

38th Conference on Neural Information Processing Systems (NeurIPS 2024).

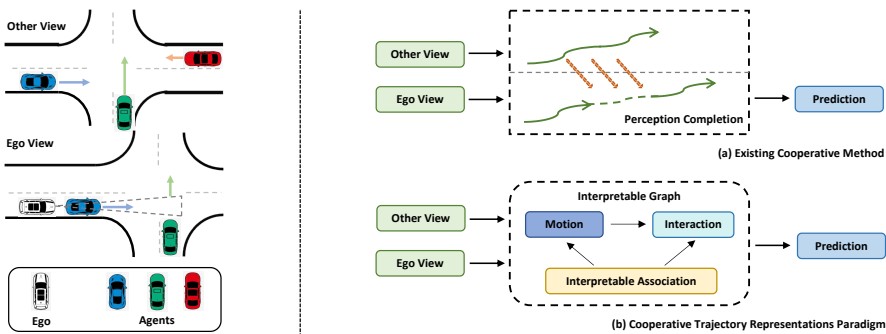

Figure 1: Scheme Comparison. (a) Existing methods utilize cooperative perception information at each frame individually then performs forecasting. (b) Our V2X-Graph considers this information from a typical forecasting perspective and employs interpretable trajectory feature fusion in an end-to-end manner, to enhance the historical representation of agents for cooperative motion forecasting.

view observations of multi-agents, the redundant data need to be leveraged interpretably. Existing research mainly focuses on single-frame feature fusion to support real-world applications and enhance detection performance [50, 44, 46, 4, 42]. A recent method [51] attempts to complement perception over the historical horizon to improve forecasting performance. These methods are depicted in fig. 1(a). However, the above single-frame methods obtain the agent state at each frame individually, which may lead to a trade-off between the states observed from distinct views, and cannot utilize motion and interaction context sufficiently, thus failing to sufficiently model the historical behavior of agents. Instead, considering historical observations from each view holistically could address these shortcomings. Compared to previous approaches, this paper explores a novel forecasting-oriented trajectory feature fusion method, which aims to enhance the historical representation of agents, including their historical motion and surrounding interactions, for motion forecasting.

To address the challenges and effectively utilize the cooperative information, we propose V2X-Graph, a graph-based framework to achieve cooperative trajectory feature fusion for motion forecasting. Theoretically, V2X-Graph offers two advantages for enhancing motion forecasting performance. (1) Forecasting-oriented cooperative representation. For accurate motion forecasting, it is common practice to represent motion features from an agent's historical trajectory and interaction features from other agents [10, 25, 58, 19]. Instead of perception complement, our method is the first to consider cooperative perception information from the typical motion forecasting perspective, which independently represents motion and interaction representations of cooperative trajectories, customized relative spatial-temporal encodings are designed to support trajectories feature fusion of each agent over historical horizon. (2) Graph-guided heterogeneous feature fusion. To support multi-agent motion forecasting in a cooperative scenario, it is essential to interpretably integrate heterogeneous motion and interaction features for each specific agent from cooperative trajectories. Drawing inspiration from graph link prediction [12, 55], a classical task that analyzes the relationship of nodes in a graph, this may further support downstream applications like node feature propagation. To achieve end-to-end optimization in V2X-Graph, we constructed a graph that represents the cooperative scenario, an interpretable association is established to guide heterogeneous feature fusion based on agent identification across views. The framework is represented in fig. 1(b).

V2X-Graph is evaluated on V2X-Seq [51], which contains vehicle-to-infrastructure (V2I) cooperative scenarios. To evaluate its effectiveness in vehicle-to-vehicle (V2V) and further more cooperative devices scenarios, we construct the first public and real-world vehicle-to-everything (V2X) motion forecasting dataset V2X-Traj. This dataset is the first to include multiple autonomous vehicles and infrastructure in every scenario, broadening the research devoted to V2X motion forecasting task. Extensive experiments conducted on V2X-Seq and V2X-Traj show the advantages of V2X-Graph in utilizing additional cooperative information to enhance the motion forecasting capability.

Our contributions are four fold: (1) We propose a forecasting-oriented representation paradigm to utilize motion and interaction features from cooperative information. (2) We design V2X-Graph, a representative framework to achieve interpretable and end-to-end trajectory feature fusion for cooperative motion forecasting. (3) We construct V2X-Traj, which is the first public and real-world dataset for V2X motion forecasting. It includes not only V2I but also V2V cooperation in every cooperative scenario. (4) Our approach achieves state-of-the-art on both V2X-Seq and V2X-Traj.

## 2   Related Work

**Cooperative Autonomous Driving.** In recent years, more and more researchers pay attention to cooperative autonomous driving, which leverages additional information from infrastructure-side devices and other vehicles to achieve system-wide performance improvement. As several public cooperative perception datasets [45, 49, 15] have been released, most of them focus on cooperative perception. Different from the single-side object detection [56, 24, 8, 47, 48], cooperative detection methods aim to promote performance and transmission latency trade-offs to support real-world applications [46, 22, 44, 41, 50, 9, 4]. Some works also dive into cooperative segmentation task [38, 43]. However, as the downstream task of cooperative perception and directly influences the actions of autonomous vehicles, cooperative motion forecasting has not been well studied. A recent endeavor supplies historical observations with perception information from infrastructure devices to improve motion forecasting performance [51]. Instead of perception completion then forecasting, this paper presents an end-to-end cooperative motion forecasting framework for cooperative trajectory feature fusion, to achieve comprehensive utilization of motion and interaction contexts from cooperative information. To further broaden the research into V2X cooperative motion forecasting, we construct the first real-world and public motion forecasting dataset for general V2X scenario, termed V2X-Traj, including not only V2I but also V2V cooperations in every scenario.

**Motion Forecasting.** Motion forecasting is an indispensable task in autonomous driving systems, which takes sequential perception results of agents and map elements into account to predict future trajectories of agents. Early works rasterize scenarios as images and deploy convolution neural networks to extract information [34, 21]. The research community turns to vectorize the representations of agents and maps for motion and interaction contexts [10]. Some works consider pooling mechanism for feature fusion [1, 14, 10, 35]. Others utilize the convolution technique to extract local features [25, 52, 5]. Inspired by the effectiveness and widespread usage of the Transformer model [36], recent works adopted attention mechanism for learning representations in motion forecasting task [26, 39, 31, 58, 33, 19, 40]. Instead of sequential perception supplementation then forecasting, V2X-Graph explores novel trajectory feature fusion to comprehensively utilize information. It represents and decouples trajectory features into motion and interaction features for each view independently, and employs customized Transformer modules for aggregating interpretable features based on agent identification, which facilitates cooperative motion forecasting.

**GNNs for Motion Forecasting.** Graph neural network (GNN) [20, 37] is a common structure for motion forecasting. A graph consists of nodes and edges, with each node typically representing information related to an agent or a map element. While edges represent the relative information between pairs of nodes. The message-passing mechanism aggregates and updates node features from their neighbors. Previous methods adopted homogeneous GNN for unified but coarse scene representation [30, 10, 23, 57, 25, 13, 11]. While recent research introduced heterogeneous GNN [54, 16] to distinguish and further extracting features based on various settings of agents [32, 29, 18, 19]. Compared to previous approaches, V2X-Graph explores leveraging heterogeneous edge encodings and interpretable graph link prediction for trajectory-based motion and interaction features fusion.

## 3   Preliminary

Cooperative motion forecasting can play an important role for autonomous driving as it better reasons the future movements of surrounding agents. The ego vehicle receives sequential perception results from cooperative devices, including infrastructures and other vehicles, to enhance the capability of motion forecasting. The contextual information in the vector map is also taken into account.

**Problem Formulation.** The inputs of cooperative motion forecasting include multiple-source trajectories and vector maps. The cooperative motion forecasting scenario is represented as $\mathcal{S} = \{\mathbf{T}, \mathbf{L}\}$, where $\mathbf{T}$ and $\mathbf{L}$ are described as follows. (1) Trajectory. In a typical V2X cooperation scenario, each cooperative device independently captures the historical status of agents as trajectories. The multi-source trajectories are denoted as $\mathbf{T} = \{\mathbf{T}_{ego}, \mathbf{T}_{other}\}$, here $\mathbf{T}_{other}$ can include received multi-source cooperative trajectories such as $\mathbf{T}_{inf}$ and $\mathbf{T}_{veh}$ from the views of infrastructure and cooperative vehicles. The total number of trajectories is $N_t = N_{ego} + N_{other}$. Trajectory information is summarized as $\mathbf{T} \in \mathbb{R}^{N_t \times T \times C_t}$, where $T$ is the historical horizon and $C_t$ is the attributes of each trajectory to depict corresponding agent (*e.g.*, tracking id, location, heading angle, detection bounding box and agent type). Specifically, the historical spatial status of each trajectory is formulated as $\{\mathbf{p}_i^t, \mathbf{r}_i^t\}_{t=1}^T$, where $\mathbf{p}_i^t \in \mathbb{R}^2$ is the trajectory $i$'s location, and $\mathbf{r}_i^t$ represents the heading theta vector at time step $t$. (2) Vector Map. Vectorized representation [10] is usually adopted for representing the

map elements in motion forecasting task, which leverages the sample points of the centerline within each lane and enables an efficient vectorized representation of spatial information. In this paper, we further consider vectorized lane segment, *i.e.*, the vector between each two neighboring sample points. The set of vectorized lane segments is denoted as: $\mathbf{L} \in \mathbb{R}^{N_l \times 2 \times C_l}$, where $N_l$ is the number of lane segments and $C_l$ is the attributes of each lane segment (*e.g.*, location and road type). The start point and the end point of the lane segment are formulated as $\{\mathbf{p}_l^{start}, \mathbf{p}_l^{end}\}$, here $l \in [1, 2, ..., N_l]$.

**Evaluation.** The output is $\mathcal{K}$ future trajectories of the specified target agent in each scenario, the best one is chosen for evaluation, here $\mathcal{K} = 6$. Evaluation metrics are minADE, minFDE and MR, standard metrics for motion forecasting. *Lower number is better*.

**Challenges.** To enhance the capability of motion forecasting considering abundant cooperative trajectories, it is essential to: (1) effectively represent the cooperative scenario, (2) efficiently utilize valuable information from redundant cooperative trajectories.

# 4 Methodology

This section presents V2X-Graph, a graph-based framework designed to achieve interpretable trajectory feature fusion for cooperative motion forecasting. To represent the cooperative scenario, it constructs a graph with node and edge encodings. To enhance cooperative trajectory feature fusion, an interpretable graph consisting of three subgraphs is designed for the aggregation of heterogeneous motion and interaction features. The overall architecture is depicted in fig. 2.

## 4.1 Scene Representation with Graph

In the graph that represents the cooperative scenario, trajectories from each view and their corresponding lane segments are independently encoded as nodes, while the relative spatial and temporal features between these nodes are encoded as edges.

**Graph Node Encodings.** We encode the node features from three perspectives: trajectory motion features, trajectory spatial-temporal features, and lane segment spatial features.

Compared to previous methods that leverage cooperative information at each frame individually, we encode differential information from each view then fuse it with correlation to mitigate the deviation caused by direct single-frame fusion. The embeddings of differential coordinates $\{\mathbf{p}_i^t - \mathbf{p}_i^{t-1}\}_{t=1}^T$ of the trajectory are considered as the motion features at each timestep. The self-attention mechanism [6] is adopted to incorporate temporal dependency. Here, missing frames are padded with learnable tokens, and the attention mechanism is enforced to only attend to the preceding time steps.

For the purposes of trajectory identification for interpretable association and motion correlation measurement, we also encode spatial-temporal features for trajectories from each view. The spatial-temporal features are encoded by incorporating the temporal dependency between the ego-centric normalized coordinates of the trajectory. Missing frames are masked in the attention module.

Overall, node encodings of trajectory from each view are formulated as:

$$\mathbf{v}_i^{mot} = \text{SelfAttn}(\text{MLP}(\mathbf{r}_i^T(\mathbf{p}_i^t - \mathbf{p}_i^{t-1})) + \text{PE}^t), \quad \mathbf{v}_i^{st} = \text{SelfAttn}(\text{MLP}(\mathbf{p}_i^t) + \text{PE}^t), \quad (1)$$

where $\text{PE}^t$ signifies the learnable positional embedding at timestep $t$, $\text{SelfAttn}(\cdot)$ is the multi-head self-attention module, and $\text{MLP}(\cdot)$ represents a multi-layer perceptron.

To enhance the representation of trajectories for future intention reasoning, the feature of the vector map structure is incorporated. Specifically, the spatial feature of lane segments are represented by the relative coordinates (from the start point to the end point of each lane segment) in an agent-centric frame [35], and further encoded as nodes for feature aggregation. The formulation is:

$$\mathbf{v}_l^{map} = \text{MLP}(\mathbf{r}_{i,l}^T(\mathbf{p}_l^{end} - \mathbf{p}_l^{start})), \quad (2)$$

where $\mathbf{r}_{i,l}^T$ is the relative heading vector between trajectory $i$ with lane segment $l$ in its current frame.

**Graph Edge Encodings.** For effective trajectory feature fusion, we design heterogeneous edge encodings, including spatial-temporal encoding and relative spatial encoding.

We use an attention module to aggregate the spatial-temporal encodings between each pair of cross-view trajectories. The spatial-temporal encoding captures the spatial-temporal correlations between two trajectories at each timestep, which facilitates motion feature fusion.

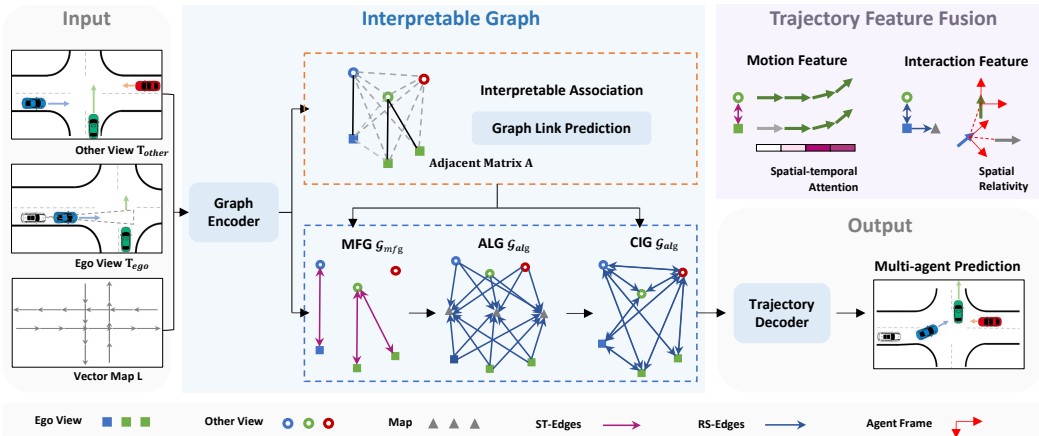

Figure 2: V2X-Graph overview. Trajectories from the ego-view and other views, along with vector map information, are encoded as nodes and edges for graph construction to represent a cooperative scenario. The novel interpretable graph provides guidance for forecasting-oriented trajectory feature fusion, including motion and interaction features. In this figure, solid rectangles represent encodings of ego-view trajectories, hollow circles represent encodings of cooperative trajectories, distinguished by distinct colors. Specifically, within the same view, the use of the same color indicates interruptions caused by occlusion. Triangles represent encodings of lane segments. In trajectory feature fusion, grey arrow indicates an missing frame in motion case, a lane segment vector in interaction case.

To capture the interaction features, we also introduce relative spatial encodings as edges.

Edge encodings can be formulated as follows:

$$\mathbf{e}_{i \to j}^{st} = \text{SelfAttn}(\text{concat}[\mathbf{v}_i^{st}, \mathbf{v}_j^{st}]), \quad \mathbf{e}_{i \to j}^{rs} = \text{MLP}(\mathbf{r}_i^T(\mathbf{p}_i - \mathbf{p}_j)), \tag{3}$$

where $\mathbf{e}_{i \to j}^{rs}$ represents both agent-agent and agent-lane interaction relations. For agent-agent interaction, the coordinates of the current frame for trajectories are denoted by $\mathbf{p}_i$ and $\mathbf{p}_j$. While for agent-lane interaction, the encoding represents the feature between current coordinate $\mathbf{p}_i$ of trajectory $i$ and the starting point coordinates $\mathbf{p}_l^{start}$ of the lane segment.

## 4.2 Feature Fusion with Interpretable Graph

To achieve comprehensive historical representations of agents in a cooperative scenario, an interpretable graph is designed for multi-view trajectory feature fusion. Serving as a guidance for heterogeneous feature representations, the Interpretable Association component (IA) establishes explicit associations between trajectories of the same agent across views. The Motion Fusion subGraph (MFG) represents cooperative motion features by considering both explicit associations and implicit spatial-temporal encodings. The Agent-Lane subGraph (ALG) fuses the features from each view with lane segment features. The Cooperative Interaction subGraph (CIG) represents dense interaction representations by leveraging the spatial encodings between different agents from all views.

---

**Algorithm 1:** Pseudo Labels Generator

**Input:** Ego-view trajectories $\mathbf{T}_{ego}$, Other-view trajectories $\mathbf{T}_{other}$
**Output:** Cross-views trajectories matching pesudo labels $\tilde{\mathcal{A}}$
**for** $t = 0$ **to** $T$ **do**
  $\mathcal{B}_{ego}, \mathcal{B}_{other} \leftarrow$ Collect ego-view, other-view detection bounding boxes from $\mathbf{T}_{ego}^t, \mathbf{T}_{other}^t$ ;
  Calculate bounding boxes IOU matrix $\mathbf{M}_t \in \mathbb{R}^{|\mathcal{B}_{ego}| \times |\mathcal{B}_{other}|}$ ;
  $\mathcal{A}^t \leftarrow$ Solving the optimal matching: Hungarian Algorithm($\mathbf{M}^t$);
  Update greedy trajectory matching at $t$: $\mathcal{A} \leftarrow \mathcal{A}^t$;

$\tilde{\mathcal{A}} \leftarrow$ Solve error matching with length intersection threshold $\epsilon_{length}$ from $\mathcal{A}$.

---

**Interpretable Association.** To achieve end-to-end optimization of heterogeneous feature fusion, we formulate the association process as a graph link prediction problem. An interpretable association component is introduced to establish interpretable associations of cross-view trajectories of agents, providing explicit guidance for the fusion of motion and interaction features. Additionally, we propose

a pseudo label generator, described in alg. 1, to collect trajectory matching labels for trajectories from the ego and another view over the historical horizon within the training set for knowledge distillation.

We denote an adjacency matrix for the two sets of trajectories $\mathbf{A} = \{a_{i,j} | i \in N_{ego}, j \in N_{other}\}$, where elements referring to associations are supervised by $\tilde{\mathcal{A}}$ which can be predicted as:

$$\mathbf{A} = \Phi_{\text{Classifier}}(\mathbf{e}_{i \to j}^{st}) \in \{0, 1\}^{|N_{ego}| \times |N_{other}|}, \tag{4}$$

here $\Phi_{\text{Classifier}}$ refers to a MLP for binary classification to determine whether there exists an association between two trajectories, thereby instructing the perception information belonging to the same agent. The classification is based on relative spatial-temporal encodings of trajectories across views.

**Motion Fusion SubGraph.** To comprehensively represent the historical motion of agents, we employ the Motion Fusion subGraph (MFG) to aggregate cooperative motion feature representations from cross-view associated trajectories. MFG models cooperative motion representations by incorporating both interpretable associations and temporal-spatial correlations.

MFG is defined as: $\mathcal{G}_{mfg} = (\mathcal{V}, \mathcal{E})$ and $\mathbf{A}_{mfg} = \mathbf{A}$, where the node set $\mathcal{V} = \{v_i | i \in N_t\}$, and the edge set $\mathcal{E} = \{e_{i,j}\}$ denotes the edges captured by $\mathbf{A}_{mfg}$. Feature fusion and update process is:

$$\mathbf{v}_i^{(k+1)} = \text{FFN}(\mathbf{v}_i^{(k)} + \text{CrossAttn}(\mathbf{v}_i^{(k)}, \mathbf{v}_j^{(k)} + \mathbf{e}_{i \to j}^{st})), \tag{5}$$

where $\mathbf{v}_i^{(k+1)}$ represents the updated feature of node $\mathbf{v}_i^{(k)}$, $\text{FFN}(\cdot)$ is a feed-forward network.

**Agent-Lane SubGraph.** We employ the Agent-Lane subGraph (ALG) to incorporate map information for cooperative motion forecasting. ALG is a bipartile graph that enables agent motion features to query relevant lane segment interaction features using relative spatial encodings. We consider all lane segments within the observation range of the current frame of agents from each view.

ALG is defined as: $\mathcal{G}_{alg} = (\mathcal{V}, \mathcal{E})$, where $\mathcal{V} = \{v_i, v_l | i \in N_t, l \in N_l\}$ and $\mathcal{E} = \{e_{i,l}\}$. The process of interaction feature aggregation and update from agents to lane segments can be formulated as:

$$\mathbf{v}_i^{(k+1)} = \text{FFN}(\mathbf{v}_i^{(k)} + \text{CrossAttn}(\mathbf{v}_i^{(k)}, \mathbf{v}_l^{(k)} + \mathbf{e}_{i \to l}^{rs} + \mathbf{a}_l)), \tag{6}$$

where $\mathbf{a}_l$ represents learnable tokens of semantic attributes associated with the corresponding lane segment, such as turn direction and road type.

**Cooperative Interaction SubGraph.** The Cooperative Interaction subGraph (CIG) is employed for cooperative interaction features representation between trajectories of distinguished agents in all views. Incorporating both interpretable associations and relative spatial correlations, CIG models denser interaction representations in a cooperative scenario.

CIG is defined as: $\mathcal{G}_{cig} = (\mathcal{V}, \mathcal{E})$ and $\mathbf{A}_{cig} = \sim \mathbf{A}$, where $\mathcal{V} = \{v_i | i \in N_t\}$, $\mathcal{E} = \{e_{i,j}\}$ denotes the set of not associated cross-view trajectories, which is captured by the adjacency matrix $\mathbf{A}_{cig}$, and all intra-view edges. The process of interaction feature fusion and update in CIG formulated as:

$$\mathbf{v}_i^{(k+1)} = \text{FFN}(\mathbf{v}_i^{(k)} + \text{CrossAttn}(\mathbf{v}_i^{(k)}, \mathbf{v}_j^{(k)} + \mathbf{e}_{i \to j}^{rs} + \mathbf{a}_{i,j})), \tag{7}$$

where $\mathbf{a}_{i,j}$ represents the learnable features of interaction attributes associated between two trajectories, such as relative headings at the current timestep. Note that CIG only considers the interaction among trajectories observed in the current frame.

## 4.3 Multimodal Future Decoder

The future motion of traffic agents is inherently multi-modal. We parameterize the distribution of future trajectories as Laplacian Mixture Model (LMM) following [58]. Every agent from each view is predicted and supervised during training phase for representations of trajectory feature fusion. While during inference phase, the decoder predicts the future trajectory of distinct agents in the cooperative scenario from the ego-view, guided by interpretable associations. Technically, we employ a MLP as a prediction head to aggregate all the intermediate features, which can be formulated as:

$$\mathbf{Z}_i^{1:T} = \text{MLP}(\text{concat}[\mathbf{v}_i^{st}, \mathbf{v}_i^{mot}, \mathbf{v}_i^{mfg}, \mathbf{v}_i^{alg}, \mathbf{v}_i^{cig}]), \tag{8}$$

where $\mathbf{Z}_i^{1:T}$ includes $\mathcal{K}$ Laplacian components $\mathcal{N}_{1:\mathcal{K}}$ with multi-modal probability distributions $p_{1:\mathcal{K}}$. The formulation for predicting the future coordinate distribution of agent $i$ at time $t$ is as:

$$P_t(o) = \sum_{k=1}^{\mathcal{K}} p_k \cdot \mathcal{N}_{1:\mathcal{K}}(\mu_x, \sigma_x, \mu_y, \sigma_y, \rho), \tag{9}$$

the future positions are generated by the center of distributions. The distribution $\mathcal{N}(\mu_x, \mu_y)$ and corresponding probability $p_k$ are generated by two MLPs separately.

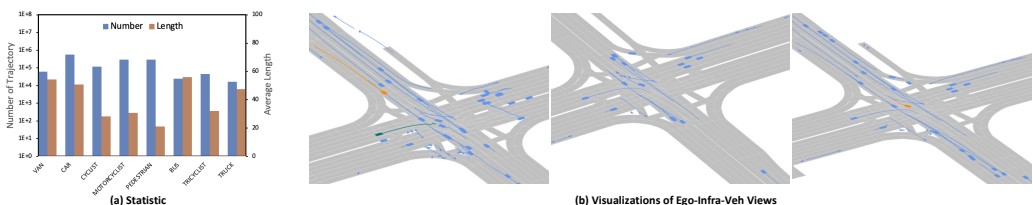

**(a) Statistic**            **(b) Visualizations of Ego-Infra-Veh Views**

Figure 3: V2X-Traj dataset. (a) Statistics of the total number and average length for the 8 classes of agents. (b) Visualizations. Orange boxes represent autonomous vehicles, blue elements denote other traffic participants and the green box denotes the target agent needs to be predicted.

## 4.4 Training Losses

To achieve interpretable trajectory feature fusion, the framework is trained in an end-to-end manner with two components of optimization objectives. The first part is the cross-entropy loss to optimize the graph link prediction. The second part includes the regression loss and classification loss to optimize the motion forecasting. Please refer to appendix for more loss details.

## 5 Experiment

In this section, we evaluate the proposed V2X-Graph framework not only on V2I cooperative scenarios but also on V2V and broader V2X cooperative scenarios.

### 5.1 Experimental Setup

**Dataset.** V2X-Graph is evaluated on both V2I and broader V2X scenarios. (1) **V2X-Seq** [51]. A public large-scale and real-world V2I dataset. V2X-Seq consists of 51,146 V2I scenarios, and each scenario is 10 seconds long with the sample rate of 10 Hz. The task is to predict the motion of agents for the next 5 seconds, given the initial 5-second observation from both infrastructure and ego-view. (2) **V2X-Traj (Ours)**. To study the effectiveness of V2X-Graph in V2V and broader V2X scenarios, especially its ability to handle more than two views of trajectories, including both V2I and V2V cooperation, we construct the first real-world and public V2X cooperative motion forecasting dataset, termed V2X-Traj. It comprises 10,102 scenarios in challenging intersections. Each scenario includes two intelligent vehicles and an infrastructure perception device. The statistics and visualization of V2X-Traj are presented in fig. 3. Each scenario lasts for 8 seconds with a sample rate of 10 Hz. The 4-second observations from each view are used to predict the future motion in the next 4 seconds. We hope the V2X-Traj dataset can facilitate the development of cooperative motion forecasting for general V2X scenarios. More details are described in the Appendix.

**Implementation Details.** For scene graph representation, V2X-Graph employs a 4-layer temporal self-attention Transformer to encode motion features, a 2-layer temporal self-attention Transformer and a 2-layer self-attention module for relative temporal-spatial feature encoding. In the interpretable graph, there are 3 layers of MFG, 1 layer of ALG and 3 layers of CIG. The dimensions of the hidden feature is set as 128, and the number of heads in all multi-head attention blocks is 16. The lane segments corresponding to agents within a observation range of 50 meters are taken into consideration. For training, the initial learning rate is set to $1 \times 10^{-3}$ and is scheduled according to cosine annealing [27]. The AdamW optimizer [28] is adopted with a weight decay of $1 \times 10^{-4}$. The model is trained for 64 epochs with batch size of 64 on a server with 8 NVIDIA RTX 4090s.

### 5.2 Main Results

**Cooperative method comparison.** We compare our method with other cooperative methods on V2X-Seq. To the best of our knowledge, PP-VIC [51] is the only existing method of the same type. Typically, PP-VIC provides the ego vehicle with infra-side perception information in a frame-by-frame manner within the historical horizon. After supplementation, the perception information is provided to popular and competitive vanilla forecasting methods DenseTNT [13] and HiVT [58]. For comparison, we also provide the output of PP-VIC to V2X-Graph in a similar way to the ego-view perception. As shown in table 1, perception completion enhances the downstream forecasting performance of HiVT and V2X-Graph. However, the trade-off in cross-view perception also leads to error propagation, resulting in performance degradation of DenseTNT. Instead, V2X-Graph enhances the historical representation of agents through trajectory feature fusion, leading to performance improvements, as evidenced by $-0.07$ in minADE, $-0.19$ in minFDE, and $-5\%$ in MR.

Table 1: Cooperative method comparison on V2X-Seq.

| Method | DenseTNT[13] | | HiVT[58] | | V2X-Graph | | |
|---|---|---|---|---|---|---|---|
| | Vehicle-only | PP-VIC[51] | Vehicle-only | PP-VIC[51] | Vehicle-only | PP-VIC[51] | Feature Fusion |
| minADE | 1.71 | 1.84 | 1.28 | 1.12 | 1.16 | 1.12 | **1.05** |
| minFDE | 2.43 | 2.56 | 2.15 | 1.97 | 2.04 | 1.98 | **1.79** |
| MR | 0.27 | 0.28 | 0.31 | 0.30 | 0.30 | 0.30 | **0.25** |

Table 2: Graph-based methods comparison on V2X-Traj.

| Method | Vehicle-only | | | V2V | | | V2I | | | V2V&I | | |
|---|---|---|---|---|---|---|---|---|---|---|---|---|
| | minADE | minFDE | MR | minADE | minFDE | MR | minADE | minFDE | MR | minADE | minFDE | MR |
| DenseTNT[13] | 1.23 | 2.09 | 0.25 | 1.20 | 2.04 | 0.25 | 1.32 | 2.34 | 0.29 | 1.26 | 2.24 | 0.28 |
| HDGT[19] | 0.91 | 1.48 | 0.14 | 0.94 | 1.57 | 0.17 | 0.94 | 1.59 | 0.16 | 0.94 | 1.56 | 0.17 |
| V2X-Graph | 0.90 | 1.56 | 0.17 | 0.77 | 1.26 | 0.12 | 0.80 | 1.30 | 0.13 | **0.72** | **1.13** | **0.11** |

**Graph-based methods comparison.** To reflect the unique advantages of V2X-Graph, we conduct evaluations on V2X-Traj and compared it with representative graph-based methods. Similar to V2X-Graph, cooperative trajectories are encoded as vanilla nodes of agents in each compared methods, for fair comparison. Experimental results in four typical settings are reported in table 2. As shown in the table, HDGT [19] achieves superior performance through precise heterogeneous design to represent relationship of agents, compared with DenseTNT [13], which employs a homogeneous graph to represent the scenario. V2X-Graph is also highly competitive in vehicle-only task, without sophisticated feature engineering and decoder design. However, our method outperforms compared methods by large margins in all cooperative settings and achieves the best performance in V2V&I co-operation, with $-0.22$ in minADE, $-0.43$ in minFDE, and $-6\%$ in MR, illustrating the effectiveness of aggregation heterogeneous motion and interaction features to enhance cooperative forecasting.

### 5.3 Ablation Study

To further illustrate the effectiveness of the method for trajectory feature fusion and the final result, we conduct ablation studies on the V2X-Traj validation set. Considering extensive ablation studies, experiments are conducted based on our small model with a hidden-size of 64.

**Effectiveness of Major Components.** Firstly, we alternately removing one of the components to illustrate the contribution of each component to the cooperative motion forecasting performance. As shown in table 3, the components within the interpretable graph separately represent the typical motion feature of historical states of agents and the interaction features with surroundings, demonstrating a marked performance enhancement of motion forecasting.

Table 3: Effect of major components.

| MFG | ALG | CIG | minADE | minFDE | MR |
|---|---|---|---|---|---|
| | ✓ | ✓ | 1.16 | 2.30 | 0.28 |
| ✓ | | ✓ | 1.26 | 2.60 | 0.30 |
| ✓ | ✓ | | 1.08 | 2.07 | 0.25 |
| ✓ | ✓ | ✓ | **0.95** | **1.79** | **0.21** |

Table 4: Effect of cooperative representations.

| MFG | ALG | CIG | minADE | minFDE | MR |
|---|---|---|---|---|---|
| | ✓ | ✓ | 1.03 | 2.04 | 0.25 |
| ✓ | | ✓ | 1.19 | 2.37 | 0.28 |
| ✓ | ✓ | | 1.08 | 2.17 | 0.27 |
| ✓ | ✓ | ✓ | **0.95** | **1.79** | **0.21** |

**Effectiveness of Cooperative Representations.** We further evaluate the effectiveness of the model in trajectory feature fusion by alternately masking the features of cooperative trajectories within each component. As demonstrated in table 4, each customized component benefits trajectory feature fusion and results in performance improvements to a certain degree. Specifically, the MFG effectively integrates the motion features of associated trajectories, leading to $-0.08$ in minADE. Relatively, the ALG and CIG components fuse the interaction features of lane segments and cooperative trajectories. These component primarily enhance the performance of long-term intention prediction, as indicated $-0.59$ in minFDE and $-7\%$ in MR.

**Effectiveness of Interpretable Feature Fusion.** Moreover, we evaluate the effectiveness of the proposed interpretable graph in aggregating heterogeneous motion and interaction features within cooperative trajectories. In table 5, the first line presents the result of the vehicle-only setting, which has no cross-view motion and interaction features fusion. We simply employ fully connections in our graph to aggregate motion and interaction features, compared with the ego-setting, there is no obvious positive effect with a large amount of features fusion. The last line shows the result of interpretable

Table 5: Effectiveness of feature fusion with interpretable graph. "Fusion Count" represents statistics average fusion counts of features per scenario. "Interpretable Fusion" indicates the aggregation of motion and interaction features through associations.

| Motion Fusion | Fusion Count in MFG | Interaction Fusion | Fusion Count in CIG | minADE | minFDE | MR |
|---|---|---|---|---|---|---|
| No Fusion | 0 | Ego | 986 | 1.06 | 2.09 | 0.27 |
| Full Fusion | 14,191 | Interpretable Fusion | 6,788 | 1.10 | 2.18 | 0.25 |
| Interpretable Fusion | 172 | Full Fusion | 6,982 | 1.08 | 2.32 | 0.27 |
| Full Fusion | 14,191 | Full Fusion | 6,982 | 1.04 | 2.09 | 0.24 |
| Interpretable Fusion | 175 | Interpretable Fusion | 6,836 | **0.95** | **1.79** | **0.21** |

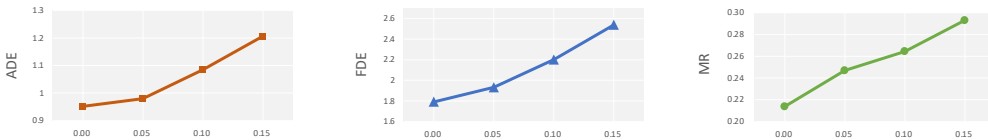

Figure 4: Effectiveness of pseudo label supervision.

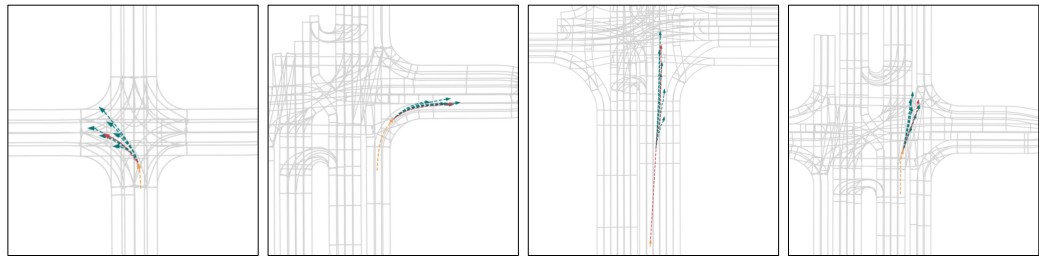

Figure 5: Qualitative results on V2X-Traj. There are several interesting cooperative scenarios at the challenging intersection, including speed-up, lane changing and turning. We visualize only the forecasting results of the target agent in each scenario for clarity. The ground-truth trajectories are shown in red, and the multimodal predicted trajectories are shown in green.

features fusion, it achieves improved performance with a low computational cost, demonstrating the effectiveness to interpretably aggregate heterogeneous cooperative features.

**Effectiveness of Pseudo Label Supervision.** To demonstrate the quality of pseudo labels and influence on the final motion forecasting performance, we disturb labels randomly for supervision and evaluate the motion forecasting results. As shown in fig. 4, as the proportion of disturbed pseudo labels increases, corresponding forecasting performance decreases, which illustrates the effectiveness of both interpretable feature fusion and pseudo label supervision. This also suggests that there is potential for further improvement in forecasting performance as the quality of labels increases.

## 6 Conclusion and Limitation

In this paper, we introduce a forecasting-oriented representation paradigm to utilize motion and interaction features from cooperative information, and present V2X-Graph, a graph-based framework to achieve interpretable and end-to-end trajectory feature fusion for cooperative motion forecasting. Comparing to existing methods that rely on single-frame perception information cooperation, our approach enhances the historical representation of agents from lightweight cooperative trajectories, and achieve improved performance in the downstream task, namely cooperative motion forecasting. Moreover, we construct V2X-Traj, the first real-world and public V2X cooperative motion forecasting dataset, expanding the research from V2I to broader V2X motion forecasting task. Experiments on both two datasets demonstrate the effectiveness of our method.

**Limitation and future work.** Compared to the single-frame method, the proposed V2X-Graph explores trajectory feature fusion, which mitigates errors from single-frame perception completion and achieves better motion and interaction representation of agents. Despite these advantages, the performance still relies on the tracking quality from each view. Jointly optimizing the performance from perception to forecasting is significant to explore in the future.

## Acknowledgment

The paper is supported by funding from Wuxi Research Institute of Applied Technologies, Tsinghua University under Grant 20242001120.

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

# Appendix

In this supplementary material, detailed information about the training loss is provided in appendix A. Comprehensive details regarding the V2X-Traj dataset are presented in appendix B. Additional experimental results are presented in appendix C. The implementation details of the compared methods on the V2X-Traj and V2X-Seq datasets are presented in appendix D. Additional qualitative results of the proposed V2X-Graph framework on the V2X-Seq dataset are provided in appendix E.

## A   Details of Training Loss

To achieve interpretable trajectory feature fusion in cooperative scenarios, the framework is trained in an end-to-end manner with two components of optimization objectives.

For the first part of optimization objectives, we utilize the cross-entropy as the knowledge distillation loss $\mathcal{L}_{dis}$ to optimize the prediction of the adjacency matrix $\mathbf{A} = \{a_{i,j}\}$, which represents the associations between trajectories of agents across views and is supervised by the pseudo labels $\alpha(\tilde{\mathcal{A}})$, here $\alpha(\cdot)$ indicates a pre-pruning strategy to mitigate the class imbalance issue. This strategy is utilized when there is no intersection between the minimum bounding rectangles of two trajectories, or when the agent types are different.

The second part includes the regression loss $\mathcal{L}_{reg}$ and classification loss $\mathcal{L}_{cls}$ to optimize the motion forecasting. For optimization, we select the trajectory with the minimum average L2 distance from the ground truth among $\mathcal{K}$ modalities. We utilize the cross-entropy loss as the selection loss to optimize $p_{i,k}$. The negative log-likelihood loss is employed as the regression loss and can be formulated as:

$$\mathcal{L}_{reg} = -\frac{1}{NH} \sum_{k=1}^{\mathcal{K}} p_{i,k} \prod_{t=T+1}^{T+H} \log \mathrm{P}(\mathbf{r}_i^T(\mathbf{p}_i^t - \mathbf{p}_i^T | \hat{\boldsymbol{\mu}}_i^t, \hat{\mathbf{b}}_i^t)), \tag{10}$$

where $\mathrm{P}(\cdot|\cdot)$ represents the probability density function of the Laplace distribution, and $\{\hat{\boldsymbol{\mu}}_i^t\}_{t=T+1}^{T+H}$, $\{\hat{\mathbf{b}}_i^t\}_{t=T+1}^{T+H}$ denote the coordinates and the corresponding uncertainties of the best-predicted trajectory for the agent. Overall, the final training loss can be formulated as follow:

$$\mathcal{L} = \mathcal{L}_{dis} + \mathcal{L}_{reg} + \mathcal{L}_{cls}. \tag{11}$$

## B   Details of V2X-Traj Dataset

V2X-Traj is the first and real-world V2X cooperative motion forecasting dataset. In this section, we provide a more comprehensive description of our V2X-Traj dataset. Dataset comparison is presented in table 6. Other details includes dataset composition (appendix B.1), collection and annotation process (appendix B.2) and additional visualizations (appendix B.3).

### B.1   Dataset Composition

V2X-Traj dataset contains a total of 10,102 scenarios, which are randomly split into the training, validation, and test set, consisting of 6,062, 2,020, and 2,020 scenarios, respectively. Each scenario comprises three independent sets of trajectories from two autonomous driving vehicles and an infrastructure-side perception device. Additionally, considering the agents are moved with the constraints of traffic rules, we also provide the static vector map and real-time traffic light signals.

**Trajectory.** Each trajectory represents the information of an agent detected and tracked independently by a single perception device. The trajectory information includes the timestamp, unique ID, agent type, location, 7-dimensional detection bounding box, heading, and velocity.

**Vector Map.** Following [3], we collect map information in the form of vectorized representations to provide valuable hints for motion forecasting. Vector maps contain lane, crosswalk, stopline, and junction elements. For each lane, we provide sample points of centerline and boundary, and semantic attributes such as turning direction, lane topology and traffic control.

**Traffic Light.** We provide real-time traffic light signals as they have a substantial impact on the behavior of traffic participants. During the data collection and storage process, we simultaneously

Table 6: Comparison with the public motion forecasting dataset. '-' denotes that the information is not provided or not available. V2X-Traj is the first cooperative dataset that supports research on V2V and broader V2V&I cooperative motion forecasting. The dataset contains abundant real-world cooperative trajectories from infrastructure and cooperative vehicles, as well as information about vector maps and real-time traffic lights.

| Dataset | Year | View | Ego-view Tracks | Infra-view Tracks | Veh-view Tracks | With Vector Map | With Traffic Light | Scenes |
|---|---|---|---|---|---|---|---|---|
| Nuscenes [2] | 2019 | Single-veh | 43 | - | - | ✓ | ✗ | 1,000 |
| Apolloscape [17] | 2019 | Single-veh | 51 | - | - | ✗ | ✗ | 103 |
| Interaction [53] | 2019 | Single-veh | - | - | - | ✗ | ✗ | 40,054 |
| Argoverse [3] | 2019 | Single-veh | 36 | - | - | ✓ | ✗ | 324,000 |
| Waymo motion [7] | 2021 | Single-veh | 73 | - | - | ✓ | ✓ | 10,4000 |
| V2X-Seq [51] | 2023 | V2I | 101 | 50 | - | ✓ | ✓ | 51,146 |
| V2X-Traj | 2024 | V2V&I | 86 | 40 | 82 | ✓ | ✓ | 10,102 |

Table 7: Detailed statistics on the total number and length of trajectories per class.

| Class | Van | Car | Cyclist | Motorcyclist | Pedestrian | Bus | Tricyclist | Truck |
|---|---|---|---|---|---|---|---|---|
| Number | 62,725 | 572,527 | 120,310 | 291,437 | 291,702 | 24,088 | 44,909 | 16,875 |
| Length | 54 | 51 | 28 | 31 | 21 | 56 | 32 | 48 |

record traffic light data at a frequency of 10 Hz. The traffic light signal information includes the timestamp, location, direction, corresponding lane ID, color status, and remaining time.

We provide detailed statistics in table 7. As the table shows, the V2X-Traj dataset contains abundant trajectories of 8 classes of agents to depict real-world V2X cooperative scenarios.

Dataset schema is represented in fig. 6. In each cooperative scenario, three sets of trajectories are independently collected by the ego vehicle, the infrastructure perception device, and another autonomous vehicle; simultaneously, data on traffic light synchronization is gathered. Additionally, we offer a comprehensive vector map covering intersections.

## B.2 Data Collection and Annotation

This subsection details the process to construct the V2X-Traj dataset.

We choose 28 urban traffic intersections in Beijing and deploy 4-6 pairs of 300-beam LiDAR and high-resolution cameras for each intersection. These infrastructure sensors can fully cover the intersection areas. We deploy one 40-beam LiDAR and six high-quality cameras for the autonomous vehicle. We provide the configuration of sensor deployment of autonomous vehicles and infrastructure in fig. 7. The perception devices of autonomous driving vehicles and infrastucture devices have trained 3D object detection and tracking models. These models are used to generate trajectory sequences.

To collect the trajectory data, the two autonomous driving vehicles were driven simultaneously and randomly through areas equipped with sensors. The V2X cooperative scenarios are collected when there is a certain overlap in the perception range of the two vehicles and the infrastructure device, resulting in the V2X cooperative trajectory sequences repository. The trajectories from each view were stored independently.

Finally, we mined interesting segments from the repository to create 10,102 cooperative scenarios. The trajectory mining process consisted of several steps, including scenario fragmentation, trajectory scoring and scenario selection. In the first step, we divided the sequences from each view into 8-second segments. In the second step, a score was assigned to each trajectory from ego-view based on the interesting behaviors of agents such as turning, speeding up, slowing down and lane changing. In the third step, we retained a total of 10,102 sequences with high-score trajectories. Within each segment, one trajectory was designated as the target agent for prediction, while the remaining trajectories were assigned as others.

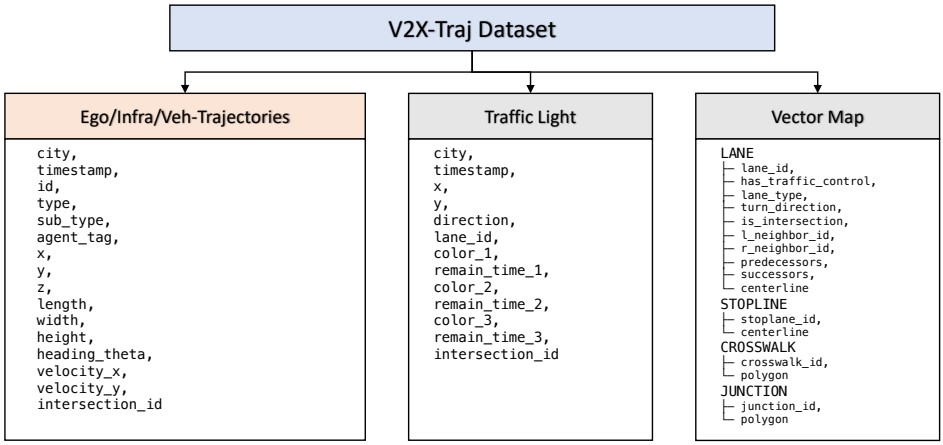

Figure 6: Schema of the V2X-Traj dataset.

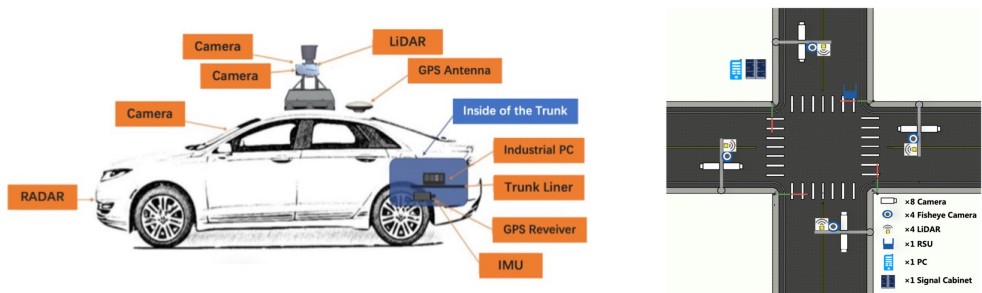

Figure 7: Sensor deployment in autonomous vehicles (left) and in infrastructure (right).

### B.3 Additional Dataset Visualization

fig. 9 presents four interesting scenarios from our V2X-Traj dataset. As the figure describes, each scenario includes trajectories from the ego-vehicle, the cooperative infrastructure and the cooperative autonomous driving vehicle. The inclusion of cooperative trajectories from both the autonomous vehicle and the infrastructure device enhances the information available to the ego-vehicle.

## C  Additional Experimental Results

### C.1  Effectiveness of Major Components

In this experiment, we conduct an additional ablation study on the V2X-Seq dataset to demonstrate the contribution of each key component to the cooperative motion forecasting performance. Similar to before, we evaluate the effectiveness by removing one of the components alternately. As shown in Table 8, each component within the interpretable graph demonstrates a marked enhancement in performance. The MFG efficiently integrates motion features from associated trajectories, leading to a significant decrease in minADE by $0.15$. The ALG incorporates interaction feature from lane segments and the CIG merges dense interaction features from cooperative trajectories, resulting in a maximum reduction of $0.51$ in minFDE and $7\%$ in MR. In summary, these components contribute significantly to performance improvements by learning heterogenerous cooperative feature representations.

| Table 8: Effect of major components. | | | | | | | Table 9: Effect of cooperative representations. | | | | | |
| --- | --- | --- | --- | --- | --- | --- | --- | --- | --- | --- | --- | --- |
| MFG | ALG | CIG | minADE | minFDE | MR | | MFG | ALG | CIG | minADE | minFDE | MR |
| | ✓ | ✓ | 1.29 | 2.11 | 0.30 | | | ✓ | ✓ | 1.29 | 2.11 | 0.30 |
| ✓ | | ✓ | 1.34 | 2.48 | 0.36 | | ✓ | | ✓ | 1.22 | 2.17 | 0.32 |
| ✓ | ✓ | | 1.30 | 2.23 | 0.34 | | ✓ | ✓ | | 1.26 | 2.11 | 0.31 |
| ✓ | ✓ | ✓ | **1.14** | **1.97** | **0.29** | | ✓ | ✓ | ✓ | **1.14** | **1.97** | **0.29** |

## C.2  Effectiveness of Cooperative Representation

We further evaluate the effectiveness of the model in cooperative feature representations on V2X-Seq by alternately masking out the infrastructure-view trajectories in each component. As demonstrated in Table 9, the customized cooperative representation learning of each component results in performance improvements to a certain degree. Concretely, the MFG effectively integrates the motion features of associated trajectories through explicit associations and implicit temporal-spatial encoding. This component leads to improved performance in motion forecasting for each future frame, as evidenced by the decrease of 0.15 in the minADE metric. Relatively, the ALG and CIG components fuse the interaction features of lane segments and cooperative trajectories through compact spatial encoding. These component primarily enhances the performance of long-term intention prediction, as indicated by a maximum reduction of 0.20 in the minFDE and 3% in the MR metrics.

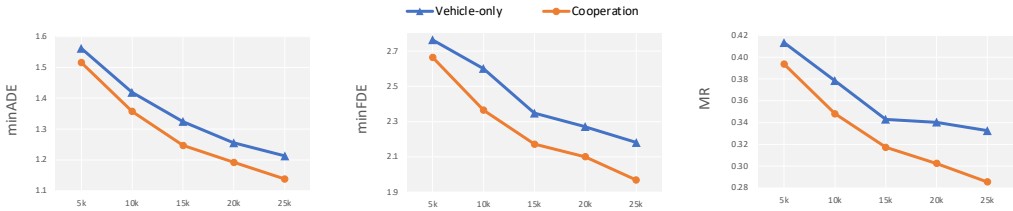

Figure 8: Scalability study on different dataset sizes.

## C.3  Scalability Study

Compared with vanilla motion forecasting, collecting large-scale data in cooperative scenarios for cooperative motion forecasting poses some realistic limitations. Therefore, it is essential to study how the performance of the method scales with dataset size.

In this experiment, we randomly split data of different sizes from the training set of V2X-Seq to train the V2X-Graph framework and evaluate the model on the full validation set. We compare the proposed V2X-Graph framework in both vehicle-only and cooperative settings to assess how the scalability of the proposed framework for cooperative motion forecasting varies with dataset size.

As shown in fig. 8, as the amount of training data increases, both vehicle-only and cooperative settings achieve improved performance. Notably, V2X-Graph achieves greater advantages in cooperation, which illustrates the scalability of the framework in cooperative motion forecasting.

## C.4  Robustness of Latency and Data Loss

We conduct robustness experiments on V2X-Seq dataset, taking both data synchronization problem and communication latency into consideration.

Specifically, we simulate latency by dropping the latest one or two frames of infra-view data during transmission. And we address the latency issue with a simple interpolation to obtain synchronized trajectory data. Experiment results in table 10 shows that there is little performance degradation to communication latency.

Data loss and sensor failures are common practical challenges. The performance advantage on real-world datasets demonstrates the robustness of our method. We further evaluate V2X-Graph under different data loss ratios on V2X-Seq dataset. Specifically, we randomly drop perception results data in each frame from the infra-view in transmission with various dropping ratios. As shown in

Table 10: Robustness of latency.

| Latency (ms) | minADE | minFDE | MR |
|---|---|---|---|
| 0 | 1.0458 | 1.7855 | 0.2527 |
| 100ms | 1.0468 | 1.7879 | 0.2562 |
| 200ms | 1.0779 | 1.8385 | 0.2688 |

Table 11: Robustness of data loss.

| Loss Ratio (%) | minADE | minFDE | MR |
|---|---|---|---|
| 0 | 1.05 | 1.79 | 0.25 |
| 10 | 1.05 | 1.81 | 0.26 |
| 30 | 1.08 | 1.85 | 0.27 |
| 50 | 1.10 | 1.88 | 0.28 |

table 11, the forecasting performance decreases as the ratio increases, but it is worth mentioning that our method outperforms the compared methods (table 1 on page 8) even under extreme conditions with a 50% loss rate.

## C.5   Parameter Size and Inference Cost Comparison

Table 12: Parameter size comparison.

| Dataset | Methods | | |
|---|---|---|---|
| V2X-Seq Param. (M) | DenseTNT [13] 1.0 | HiVT [58] 2.6 | V2X-Graph 5.0 |
| V2X-Traj Param. (M) | DenseTNT [13] 1.0 | HDGT [19] 12.1 | V2X-Graph 4.9 |

As shown in table 12, the parameter size of V2X-Graph is comparable with other forecasting methods.

Table 13: Inference cost comparison.

| Dataset | Methods | | |
|---|---|---|---|
| V2X-Seq Latency (ms) | PP-VIC [51] + DenseTNT [13] 161.45 + 371.38 | PP-VIC [51] + HiVT [58] 161.45 + 53.30 | V2X-Graph 51.50 |
| V2X-Traj Latency (ms) | DenseTNT [13] 168.88 | HDGT [19] 1260.70 | V2X-Graph 52.69 |

Then we conduct the inference experiment on single NVIDIA GTX 4090 and compare the inference cost. As for the experiment results shown in table 13, the proposed V2X-Graph is even faster than the compared vanilla motion forecasting methods, benefiting from the synchronous temporal state modeling and integration.

## D   Additional Implementation Details

In this section, we provide implementation details of compared methods in our experiemnts.

**Implementation Details of HiVT.** We compare our proposed V2X-Graph with the official evaluation of the V2X-Seq dataset [51] from https://github.com/AIR-THU/DAIR-V2X-Seq. For fair comparison, we re-implemented a larger model of HiVT [58] with a hidden size of 128 for improved performance as reported in the corresponding paper.

**Implementation Details of DenseTNT.** For comparison, we re-implemented the classical homogeneous graph method DenseTNT [13] on our V2X-Traj dataset using their official code package, from https://github.com/Tsinghua-MARS-Lab/DenseTNT. The model is trained using the default settings of two stages on the V2X-Traj training set, utilizing a server with 8 NVIDIA RTX 3090s. During the first stage, all modules are trained, except for the goal set predictor, for 16 epochs. In the second stage, the goal set predictor is trained for 6 epochs. The batch size is set to 64, the initial learning rate is 0.001, and it decays by 30% every epoch. The hidden size of the feature vectors is set to 128, and the head number of our goal set predictor is 12.

**Implementation Details of HDGT.** We further re-implemented the advanced heterogeneous graph method HDGT [19] based on their official code from https://github.com/OpenDriveLab/HDGT. We

re-implemented a larger model of HDGT with a hidden size of 256, and the number of heads in all multi-head attention blocks is 64, for better performance for comparison. The size of the kernel of the AgentTemporalEncoder in HDGT is set to 40-13-3 to accommodate the specific observation horizon of 40 time steps in V2X-Traj. During training phase, we follow the default settings, which uses the AdamW optimizer with an initial learning rate of $5 \times 10^{-4}$, weight decay of $1 \times 10^{-4}$, and batch size of 64. For the V2X-Traj dataset, the number of training epochs is set to 30, with a 1-epoch warmup and linear decay to 0. The type-specific agent distance buffer hyperparameters are empirically set to 30 meters for vehicles, 10 meters for pedestrians, and 20 meters for cyclists.

# E  Additional Qualitative Results

In this section, we present qualitative results on V2X-Seq dataset, including the visualizations of interpretable association (appendix E.1) and the visualizations of our proposed V2X-Graph and compared method (appendix E.2).

## E.1  Visualizations of Interpretable Association

Here we present the visualizations of the interpretable association on V2X-Seq. For clarity, we only visualize the associations of the historical trajectories of the target agent from the ego-view (fig. 10 (a)) and the infrastructure-side (fig. 10 (b)). Dashed circles indicate a part of the additional information from infrastructure-side trajectories that can be utilized. As shown in the figure, our method enables the interpretable association of trajectories across views, serving as guidance for the end-to-end learning of cooperative trajectory representations.

## E.2  Qualitative Results on V2X-Seq

We present additional qualitative results on the V2X-Seq dataset; three challenge scenarios are selected for methods comparison. In particular, the motion forecasting results of HiVT [58], which employs PP-VIC [51] to utilize cooperative information, are shown in fig. 10 (c), and the results of our proposed V2X-Graph are shown in fig. 10 (d). In the methods comparison, our method exhibits exceptional performance in motion forecasting, in particular of predicting long-range intentions, which demonstrates the ability of proposed V2X-Graph for further utilization of information within cooperative trajectories.

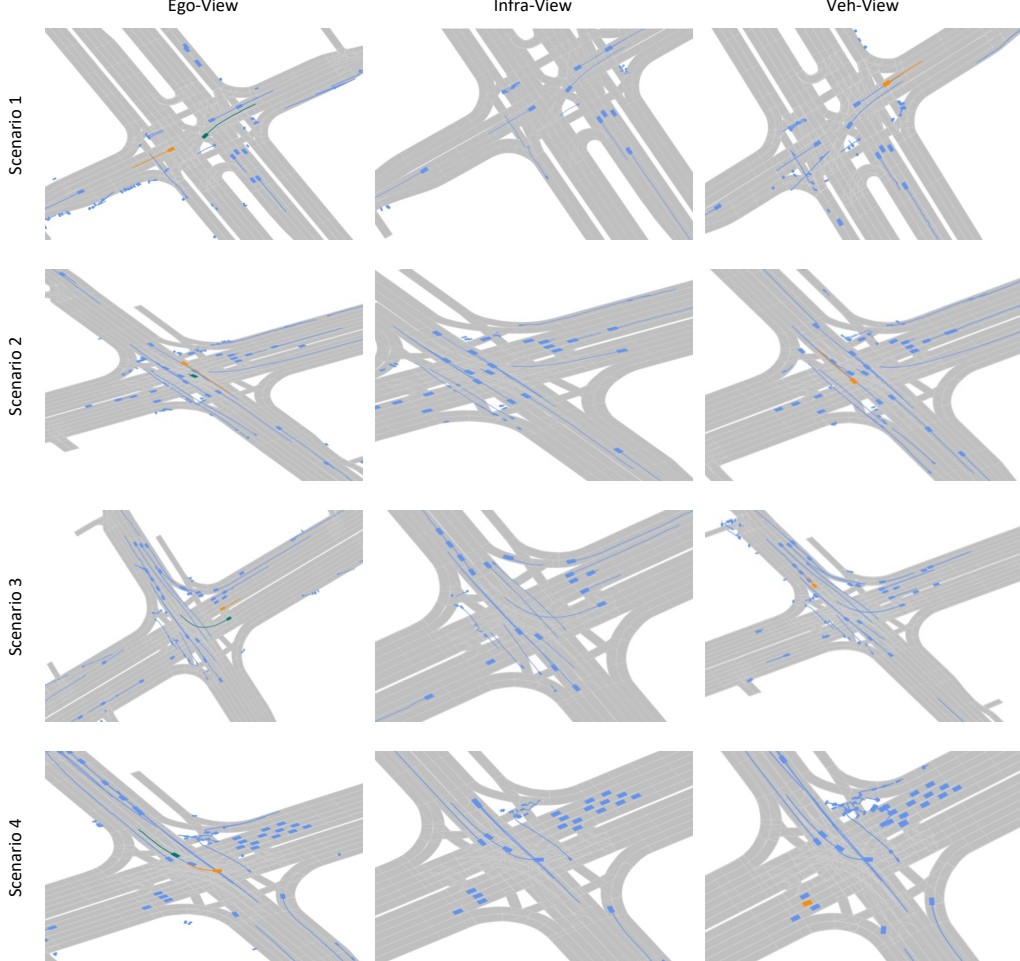

Figure 9: Visualizations of the V2X-Traj dataset. Each scenario consists trajectories from the ego-vehicle, the cooperative infrastructure and the cooperative autonomous vehicle. In this figure, orange boxes represent autonomous vehicles, blue elements denote traffic participants, and green boxes denote the target agent that needs to be predicted.

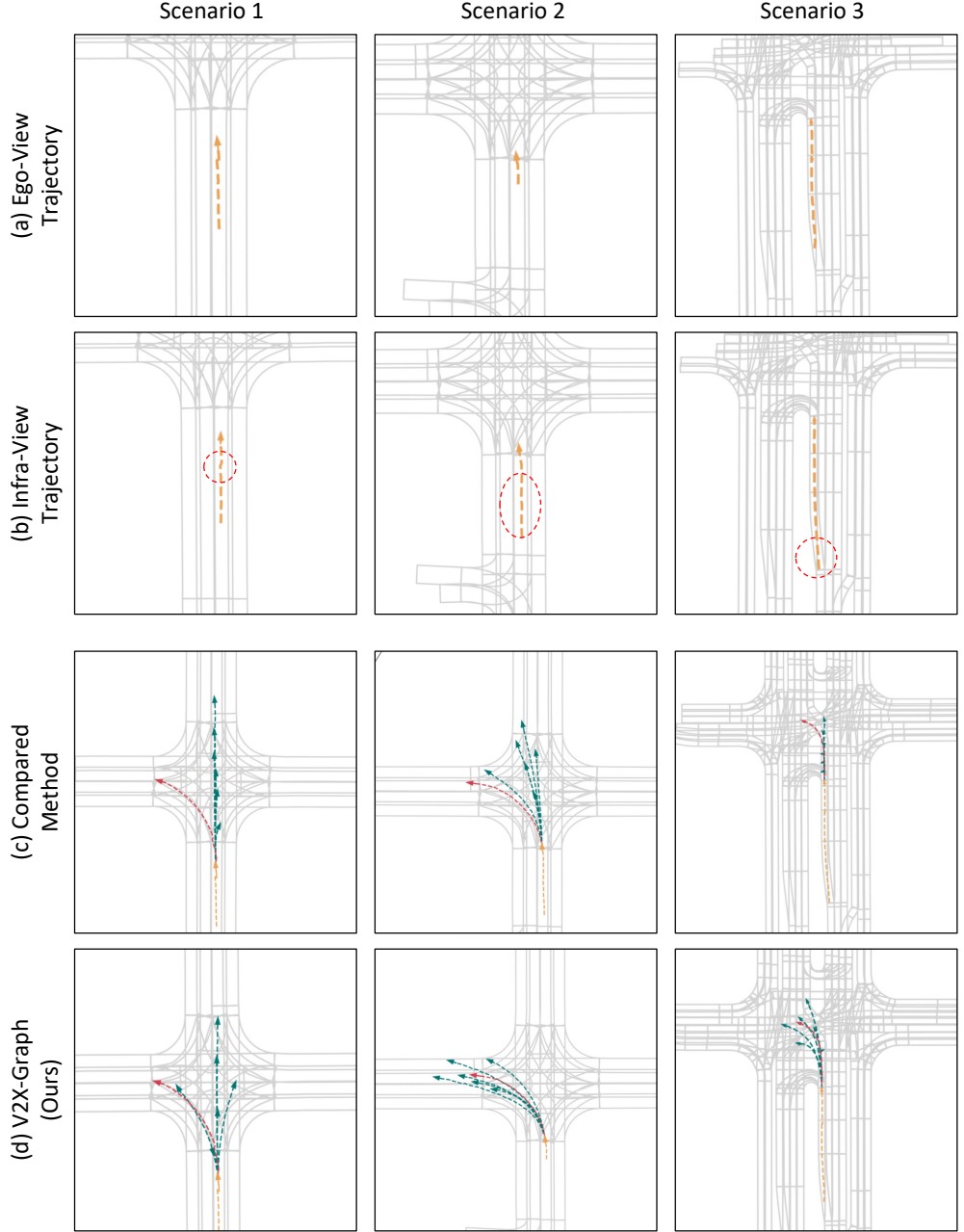

Figure 10: Qualitative results on three challenge scenarios over V2X-Seq. The historical trajectories of the target agent are shown in yellow. The red dashed circles indicate a part of the enhanced information from the infrastructure view. The ground-truth trajectories are shown in red, and the predicted trajectories are shown in green.

