# OpenReview forum: "Learning Cooperative Trajectory Representations for Motion Forecasting"
_NeurIPS.cc/2024/Conference — NeurIPS 2024 poster_

### Official Review · Reviewer_w4tS · 2024-07-05

**Soundness:** 2
**Presentation:** 3
**Contribution:** 2
**Rating:** 5
**Confidence:** 4

**Summary:**

This paper introduces V2X-Graph, a novel framework using trajectories of agents (include ego and others) and vector map as inputs for cooperative motion forecasting that fuses trajectory features in an interpretable, end-to-end manner. The authors evaluate V2X-Graph using V2X-Seq, a vehicle-to-infrastructure (V2I) motion forecasting dataset. Additionally, they create V2X-Traj, the first real-world dataset for vehicle-to-everything (V2X) motion forecasting, including scenarios with multiple autonomous vehicles and infrastructure. Author shows the V2X-Graph is achieving state-of-the-art results on both V2X-Seq and V2X-Traj datasets.

**Strengths:**

1) The paper is well-written, with a clear introduction of methods and detailed explanations of the experiments conducted.
2) This paper proposes a novel framework to address motion forecasting in V2X scenarios that not only encodes multiple trajectories with map information but also designs additional graphs for heterogeneous feature representations. Moreover, the paper discusses the effectiveness of different components.
3) This paper introduces the V2X-Traj dataset, which includes rich information such as trajectories, traffic lights, and maps, making it a valuable resource for the community to study V2X scenarios.

**Weaknesses:**

The proposed method emphasizes V2X; however, the inputs do not include important factors such as traffic lights, lane markings, and pedestrians, which may significantly impact the future motion of the vehicle. There is concern that V2X-Graph oversimplifies the problem.

**Questions:**

1. There are more works of GNNs for motion forecasting such as [1][2][3].
2. What’s the efficiency or latency of this framework given it has quite complicated network components?
3. Please check the weakness.

[1] Mohamed, Abduallah, et al. "Social-stgcnn: A social spatio-temporal graph convolutional neural network for human trajectory prediction." Proceedings of the IEEE/CVF conference on computer vision and pattern recognition. 2020.

[2] Li, Jiachen, et al. "Evolvegraph: Multi-agent trajectory prediction with dynamic relational reasoning." Advances in neural information processing systems 33 (2020): 19783-19794.

[3] Girase, Harshayu, et al. "Loki: Long term and key intentions for trajectory prediction." Proceedings of the IEEE/CVF International Conference on Computer Vision. 2021.

**Limitations:**

The paper discusses the limitations of the work in section 6.

---

> ### Author Rebuttal · Authors · 2024-08-07
>
> Dear Reviewer w4tS: \
> Thanks for your valuable feedback on our work.
> We have carefully considered your suggestions and would like to respond to each of your main comments regarding our weaknesses and questions.
>
> **W1.**
> **The structure and information of the input data for V2X-Graph primarily follow popular motion forecasting methods**, such as [1][2][3], which involve the geometry and semantic attributes of lanes, and all types of agents in the scenario, including vehicles, bicycles, and pedestrians.
> Pedestrian trajectories are also modeled and predicted. We believe this formulation of the problem is representative.
>
> Our method focuses mainly on V2X.
> The inclusion of other factors, such as traffic lights, could benefit our approach and is worth exploring in the future.
>
> We apologize for any potential misunderstandings in our paper and will detail these factors in Section 3 and Figures 1-2 in the revised version for clarity.
>
> **Q1.**
> Thanks for reminding us of these excellent works on GNNs for motion forecasting.
> These works are insightful in leveraging GNNs for modeling interactions between agents and their surroundings.
> We will include these papers as related works in the next version.
>
> **Compared with these works, we introduce a unique design of GNN to interpretably integrate motion and interaction features from cooperative data for motion forecasting.**
>
>
> **Q2.**
> Inference efficiency is important for practical application.
> We further report the parameters and the inference cost in Subsection 2.3 of General Rebuttal.
> **The experimental results demonstrate the inference efficiency of V2X-Graph in complex cooperative scenarios.**
>
> [1] Gao et al. Vectornet: Encoding hd maps and agent dynamics from vectorized representation. CVPR 2020. \
> [2] Shi et al. Motion transformer with global intention localization and local movement refinement. NeurIPS 2022. \
> [3] Zhou et al. Hivt: Hierarchical vector transformer for multi-agent motion prediction. CVPR 2022.

---

> > ### Comment · Reviewer_w4tS · 2024-08-12
> >
> > Thank you for your comprehensive reply and experimental results.

---

> > > ### Author Response · Authors · 2024-08-13
> > > **Official Comment by Authors**
> > >
> > > Dear Reviewer w4tS:
> > >
> > > Thanks for your response to our rebuttal.
> > >
> > > In the rebuttal, we primarily made clarifications to potential misunderstandings regarding the formulation of the problem and provided experimental results addressing your concerns about computational efficiency.
> > >
> > > We are very glad that our response and experimental results have addressed your main concerns and meet your expectation.
> > >
> > > On that basis, will you raise your rating, or is there any other concern that needs further clarification? We look forward to your ratification and further valuable discussions :).
> > >
> > > Best Regards,\
> > > 4569 Authors.

---

### Official Review · Reviewer_vBSp · 2024-07-05

**Soundness:** 2
**Presentation:** 2
**Contribution:** 3
**Rating:** 5
**Confidence:** 3

**Summary:**

This paper presents the V2X-Graph method for cooperative motion forecasting. In a cooperative autonomous driving setting, an autonomous driving vehicle receives sensor data from surrounding vehicles and infrastructure-side devices.

Existing cooperative autonomous driving approaches focus on perception completion, where the vehicles receive only perception data, and motion forecasting is performed with completed perception data without considering the cooperative setting.

The proposed V2X-Graph method is a graph-based framework that performs cooperative motion forecasting with trajectory feature fusion. It fuses history trajectories from different views with an Interpretable Graph. The Interpretable Graph consistents of three subgraphs: Motion Fusion subGraph (MFG), Agent-Lane subGraph (ALG), and Cooperative Interaction subGraph (CIG).

The authors also provide a new dataset V2X-Traj. Compared to the existing V2X-Seq dataset which contains only vehicle-to-infrastructure (V2I) cooperative scenarios, V2X-Traj also includes vehicle-to-vehicle (V2I) cooperative scenarios, making it vehicle-to-everything (V2X).

The authors evaluated their V2X-Graph method on the existing V2X-Seq dataset as well as their new V2X-Traj dataset. They compared against the PP-VIC baseline. The result shows that adding trajectory feature fusion improves the performance over the PP-VIC baseline.

**Strengths:**

* The proposed V2X-Graph method improves the cooperative motion forecasting performance over the PP-VIC baseline.

* The new V2X-Traj dataset will be very useful to the community.

**Weaknesses:**

* I am not fully convinced by the motivation of why perception completion is not enough for cooperative motion forecasting. It will be useful to give some examples to explain how feature fusion helps.

* The writing can be improved. For example, it will be useful to give an overview of the architecture either in the caption of Figure 2 or at the beginning of Section 4. The module names MFG, ALG, CIG, etc in the architecture diagram are not explained until Section 4.2. The evaluation setup of Table 2 (Graph-based methods comparison) is not clearly explained.

**Questions:**

* Table 1: Why are the PP-VIC + HiVT results different from those in the original PP-VIC paper?

* Line 284: How are the observations of the same agent from different views aggregated?

**Limitations:**

Yes

---

> ### Author Rebuttal · Authors · 2024-08-07
>
> Dear Reviewer vBSp: \
> Thank you for your valuable feedback on our work.
> We have carefully considered your suggestions and would like to respond to each of your main comments regarding our weaknesses and questions.
>
> **W1.**
> Instead of single-frame perception completion method, the proposed trajectory-based feature fusion have advantages in two perspectives.
>
> First, the observation of the agents in different views could be different due to various sensor perspectives and configurations; direct fusion may lead to deviations. Instead, utilizing differential information from each view and integrating it by correlation for forecasting may mitigate the effect.
>
> Second, given the aforementioned shortcomings, perception completion methods obtain the agent state at each frame separately, lacking modeling across historical timesteps. This deficiency negatively impacts motion and interaction feature modeling in downstream forecasting.
>
> **W2.**
> Thanks for your suggestion.
> We will give an overview of the architecture and briefly introduce the key modules at the beginning of Section 4, and elaborate on the evaluation setup in Table 2 for clarity and accessibility.
> We will address other similar issues in the revised version.
>
> **Q1.**
> In the original paper, HiVT is evaluated with a hidden size of 64.
> We follow the updated evaluation in that paper, using a hidden size of 128 to achieve better performance of HiVT for a fair comparison.
>
> **Q2.**
> We compare the proposed V2X-Graph with popular and effective graph-based methods in cooperative scenarios.
>
> Both V2X-Graph and the compared graph-based methods encode trajectories from each view as different nodes.
> The methods are trained in cooperative scenarios to integrate cooperative data for performance improvement.
>
> For the compared graph-based methods, the feature integration among agents observed from all views follows the original settings, including full edge connections and edge encodings similar to those in vanilla forecasting tasks.
> In contrast, for the proposed V2X-Graph, **we design heterogeneous edge encodings to integrate motion and interaction features from cooperative data, with the integration guided by graph link prediction**.
>
> Graph-based comparisons across various cooperative scenarios demonstrate the effectiveness and advantages of the proposed method.

---

> > ### Comment · Reviewer_vBSp · 2024-08-08
> > **Thank you for your response**
> >
> > Thank you for your response. I will keep my rating.

---

> > > ### Author Response · Authors · 2024-08-13
> > > **Official Comment by Authors**
> > >
> > > Dear Reviewer vBSp:
> > >
> > > We sincerely appreciate your recognition and support for our work.
> > >
> > > Best Regards,\
> > > 4569 Authors.

---

### Official Review · Reviewer_vdKw · 2024-07-13

**Soundness:** 2
**Presentation:** 3
**Contribution:** 2
**Rating:** 5
**Confidence:** 2

**Summary:**

This paper tackles the cooperative motion forecasting problem for vehicles. This paper introduces additional information other than the ego view agent from other view to expand the perception field of the prediction. The authors propose a graph network to extract information for multimodal trajectory prediction. The authors introduce a new dataset, namely V2X-Traj, for this problem setting. Experiments show the efficacy of the proposed method.

**Strengths:**

1. The problem setting is new. With more and more autonomous vehicles on the road, it is possible to include cooperative information.
2. The experimental results on both datasets show that V2X-Graph outperforms existing methods.

**Weaknesses:**

1. The cooperative setting is limited to two vehicles.
2. The scalability of the V2X-Graph framework in terms of computational resources and its performance with larger datasets or in more complex environments is not extensively discussed. The training cost and inference cost should be compared with the baselines.
3. There is not error analysis. When does the method/cooperative setting fail and why?
4. Potential cooperative forecasting problems are not considered. How does the V2X-Graph framework handle communication delays or data loss in real-time applications?

**Questions:**

Please address the weaknesses.

---Post rebuttal
Thank you for the author's response. The reply addressed most of my concerns and the added experiments on inference efficiency are important. I will increase my score

**Limitations:**

The authors have discussed the potential limitations.

---

> ### Author Rebuttal · Authors · 2024-08-07
>
> Dear Reviewer vdKw: \
> Thanks for providing valuable feedback on our work.
> We will address each of the limitations you have pointed out in your comments.
>
> **W1.**
> Yes, **we focus more on the representative scenario unit at the current stage, which involves two vehicles and one roadside device. It can represent the common cooperative scenarios, including vehicle-to-vehicle (V2V) and vehicle-to-infrastructure (V2I).**
>
> In this setting, we evaluate the proposed method with real-world dataset, which reflects the real complex traffic situation.
> **The experimental results demonstrate that the proposed V2X-Graph can be expanded from two views to three views (see Table 2 on page 8). Theoretically, the method can also be directly expanded to more than three views.**
>
> To collect the dataset, we allow multiple autonomous vehicles to drive independently and collect data on cooperative scenarios with specific perception coverage.
> The V2V scenarios with more than two vehicles are limited due to data distribution.
> **As the data volume accumulates, more diverse cooperative scenarios will be introduced in the following works.**
>
> **W2.**
> Thanks for your suggestions. We further report the model parameters and inference cost in Subsection 2.3 of General Rebuttal.
>
> V2X-Seq is a large-scale challenging cooperative dataset, and V2X-Traj has more complex scenarios.
> **It can be seen from the experimental result that the proposed V2X-Graph achieves a better performance-efficiency balance.**
>
> The lack of public dataset hinders the further development of cooperative motion forecasting. We release a larger and more challenging dataset, i.e., V2X-Traj, but a more complex one is needed.
> We are going to enlarge the dataset and introduce more complex scenarios.
>
> **W3.**
> Transimission latency and data loss are two common challenges in practice. We add error analysis in Subsections 2.1 and 2.2 of General Rebuttal.
>
> Under non-ideal communication conditions, performance is inevitably influenced.
> However, **the effect of errors in several frames is limited since V2X-Graph models cooperative data holistically, and the framework still outperforms the compared methods.**
>
> **W4.**
> We report the performance when transimission latency and data loss occurs in Subsections 2.1 and 2.2 of General Rebuttal.
> The experimental results demonstrate the effectiveness in practical applications.
> Thanks for your valuable suggestion, more disscusion will be added in the revision paper.

---

### Official Review · Reviewer_zJGd · 2024-07-14

**Soundness:** 2
**Presentation:** 3
**Contribution:** 3
**Rating:** 5
**Confidence:** 4

**Summary:**

The paper introduces a novel graph-based framework called V2X-Graph for learning cooperative trajectory representations in motion forecasting for autonomous vehicles. V2X-Graph aims to enhance the motion prediction capabilities of autonomous vehicles by leveraging cooperative information from vehicles and traffic infrastructure. The framework represents trajectories as nodes in a graph and fuses motion and interaction features from multiple views to achieve interpretable end-to-end trajectory feature fusion. The authors also constructed the first real-world V2X motion forecasting dataset, V2X-Traj, which includes cooperative scenarios with multiple autonomous vehicles and traffic infrastructure. Experiments conducted on both the V2X-Seq and V2X-Traj datasets demonstrate the advantages of V2X-Graph in utilizing additional cooperative information to enhance motion forecasting capabilities.The creation of the V2X-Traj dataset, a real-world, public dataset for V2X motion forecasting, adds significant value to the field, promoting further research in this domain.

**Strengths:**

- The paper introduces a novel forecasting-oriented representation paradigm, V2X-Graph, which leverages cooperative perception information from multiple sources (e.g., infrastructure and other vehicles) to enhance motion forecasting for autonomous driving. This approach is unique in focusing on the holistic historical motion and interaction features rather than single-frame perception.
- The methodology section (pages 4-6) is detailed and provides a comprehensive description of the V2X-Graph framework, including scene representation with graphs and trajectory feature fusion. The use of graph neural networks (GNNs) to encode cooperative scenarios and perform motion forecasting is well-justified.
- The paper includes extensive experimental results (pages 8-10) on both the V2X-Seq and V2X-Traj datasets, demonstrating the effectiveness of the proposed approach with clear performance improvements over baseline methods.
- The paper is well-structured, with a clear flow from the introduction of the problem (pages 1-2), through the related work (pages 2-3), to the proposed methodology and experimental results. Each section builds logically on the previous one, making the paper easy to follow. Figures and tables (e.g., Figure 1 on page 3 and Table 1 on page 8) are effectively used to illustrate key concepts and results, enhancing the reader’s understanding.
- By addressing the underutilization of cooperative perception information in motion forecasting, the paper tackles a critical challenge in autonomous driving. The proposed V2X-Graph framework has the potential to significantly improve the safety and efficiency of autonomous vehicles by leveraging a richer set of perception data.

**Weaknesses:**

- While the application of graph neural networks (GNNs) to encode cooperative scenarios is well-executed, the approach may lack significant novelty. GNNs have been widely applied in various fields, including autonomous driving. The paper would benefit from a more detailed comparison with existing GNN-based methods for motion forecasting to highlight the unique contributions and advancements of the proposed V2X-Graph framework (page 6, Section 3.2).
- The experimental evaluation compares the proposed method primarily against a few baselines. Including a broader range of state-of-the-art methods for motion forecasting, especially those utilizing cooperative perception, would strengthen the validation of the proposed approach. This would provide a more comprehensive view of where V2X-Graph stands in the context of current research.
- Limited Discussion on Real-World Implementation: lacking a thorough discussion on the practical challenges and considerations of implementing the V2X-Graph framework in real-world autonomous driving systems. Issues such as communication latency, data synchronization, and robustness to sensor failures are crucial for practical deployment but are not sufficiently addressed (page 10, Section 5, Discussion).
- Some parts of the methodology section are dense and might be challenging for readers to follow, particularly those less familiar with GNNs. Simplifying the mathematical notation or providing additional explanatory text and illustrative examples would improve clarity and accessibility

**Questions:**

While the experiments demonstrate the effectiveness of V2X-Graph on the V2X-Seq and V2X-Traj datasets, these datasets may not capture the full range of scenarios encountered in real-world driving. Expanding the evaluation to include more diverse and challenging driving environments would enhance the generalizability and robustness of the proposed method.
V2X-Traj dataset, while a valuable contribution, might have inherent biases that are not discussed in the paper. Providing a more detailed analysis of the dataset's composition, including the diversity of scenarios and the representativeness of different driving conditions, would be beneficial (page 7, Section 4.1).

**Limitations:**

- The paper briefly mentions some limitations in the discussion section, but it does not provide an in-depth analysis. The authors should more thoroughly acknowledge the limitations of their approach, including potential biases in the V2X-Traj dataset and the constraints of their methodology. Specific limitations such as the scalability of the V2X-Graph framework in highly dynamic and dense traffic environments, potential issues with data communication latency, and the synchronization of information from multiple sources are not adequately discussed. Including a more detailed assessment of these factors would be beneficial.

Others:
- The paper should address practical challenges such as the computational overhead required for real-time processing, the robustness of the system in adverse weather conditions, and the reliability of V2X communication in various urban and rural environments.
Discussing potential solutions or future work aimed at overcoming these challenges would strengthen the practical applicability of the proposed method.

- While the paper makes a significant contribution to the field of cooperative motion forecasting, it would benefit from a more thorough discussion of its limitations and potential negative societal impacts. By addressing these aspects in detail and providing constructive suggestions for mitigating these issues, the authors can enhance the robustness, and applicability of their work.

---

> ### Author Rebuttal · Authors · 2024-08-07
>
> Dear Reviewer zJGd: \
> Thanks for your thorough review and valuable suggestions on our work.
> We have carefully considered your suggestions and would like to respond to each of your main comments regarding our weaknesses and questions.
>
> **W1.**
> Graph neural networks (GNNs) are common practice for the vanilla motion forecasting task.
> In this task, GNNs with full edge connections can effectively model the interaction of the specific agent and its surroundings.
> Early works adopted homogeneous GNNs, while recent research has introduced heterogeneous GNNs to address the assumption of varying patterns in different interaction cases (e.g., vehicle to lane, vehicle to pedestrian).
>
> While considering the cooperative motion forecasting task, the challenges rised: how to formulate the fusion paradigm to effectively utilize the trajectory data of each agent from each view?
>
> **Existing graph-based methods cannot leverage cooperative information effectively (Table 2 on page 8).** To address the challenges, **we explored the unique cooperative trajectory representation paradigm through a novel design of GNN**, which integrates motion and interaction features via heterogeneous edge encodings, and further introduces graph link prediction in this task to guide interpretable fusion.
>
> **W2.**
> Most of the existing works in cooperative autonomous driving community pay more attention on cooperative perception, especially the single-frame cross-view feature fusion methods.
>
> While considering the communication latency chanllenge in cooperative motion forecasting task, we focus on lightweight cooperative information, namely perception results, instead of raw sensor data or features.
>
> **PP-VIC stands out as the only existing method based on perception results and the representative method of the single-frame perception approaches for the downstream forecasting task.**
> It analyzes multi-view perception relations along trajectories to achieve completion.
> **Our V2X-Graph achieve a better performance compared with it.**
>
> Thanks for your suggestion, we will give more discussion in the revision paper.
>
> **W3.**
> Thanks for pointing it out, we further report the experimental results in the Subsections 2.1-2.2 of General Rebuttal.
> Considering cooperative data holistically, V2X-Graph shows robustness against errors in local frames due to latency or data loss.
>
> **W4.**
> Thanks for your suggestion, we will simplify the representations in the methodology section for clarity in the revision paper.
>
> **Q.**
> Thanks for your valuable question. **We report more details of the proposed V2X-Traj dataset in Section 3 of General Rebuttal.**
>
> We allow multiple autonomous vehicles to drive independently and collect data on cooperative scenarios with specific perception coverage.
> **To ensure the diversity of traffic and weather conditions, the V2X-Traj dataset is collected from several challenging and representative urban intersections, and the data is collected along the whole year.**
>
> However, the collected scenarios are limited by the roadside sensor installation, which is mostly at the intersections. We are continuely collecting the data and will provide more diverse cooperative scenario data in the future, such as the scenarios with multiple vehicles in corridor scenes.
>
> **L1.**
> Thanks for your suggestion, we will expand our discussion in the revision paper.
>
> Compared with the single-frame method, the proposed V2X-Graph explores trajectory feature fusion, which mitigates errors from single-frame perception completion and offers additional advantages such as better motion and interaction representation of agents.
>
> Although V2X-Graph has the above advantages, the performance still relies on tracking quality from each view.
> A tracking error could introduce noises in trajectory representation and association.
>
> To jointly optimize the performance from perception to forecasting, we are going to further explore the end-to-end cooperative forecasting.
>
> **L2.**
> The inference cost and the robustness under non-ideal communication condition are the two main concerns for practical applications.
>
> We further report the experimental results in Section 2 of General Rebuttal. **It can be seen that the inference cost is acceptable and the performance degradation is limited in poor communication conditions.**
>
> As for the impacts of weather conditions, it will directly influence the perception performance, and may further introduce input noise to our V2X-Graph.
> Expanding the framework to the end-to-end cooperative forecasting is a possible solution to address the challenge, and we are working on that as a future work.
>
> **L3.**
> Thanks for your suggestion, we will provide additional discussion on potential negative societal impacts in the revised version.
>
> With more and more autonomous vehicles on the road, it is becoming possible to share information among vehicles and infrastructure.
> However, there are potential negative societal issues such as privacy concerns and backdoor attacks.
> Exploring security encryption and anomaly detection for shared cooperative information will benefit research in this field.

---

> > ### Comment · Reviewer_zJGd · 2024-08-11
> > **Response**
> >
> > Thank you for your detailed rebuttal and for addressing my previous comments. While I appreciate the efforts taken to differentiate your work from existing approaches, I still have concerns about the novelty of the GCN approach used in the paper. To better understand the contributions and improvements over existing GCN methods in the motion prediction domain, I would like to request the following:
> >
> > - Comparison with Existing GCN Methods. Please provide a detailed comparison of your GCN approach with other GCN methods specifically used in the motion prediction domain. Highlight the specific differences in the architecture, edge encoding, and feature propagation mechanisms. This comparison should include a discussion on how these differences lead to improved performance or new capabilities.
> > - Improvements Over Existing Methods. Discuss the specific improvements your GCN method brings over existing GCN methods in terms of accuracy, interpretability, and computational efficiency. Include quantitative metrics or qualitative analyses that support these claims.
> > - Tailored Design for V2X-Traj Dataset. Given the unique characteristics of the V2X-Traj dataset, I would like to know if the GCN architecture was tailored to leverage the specific properties of this dataset. For example, does the GCN architecture take into account the unique cooperative nature of the data and the varying levels of interaction between agents? If so, please describe these design choices and how they contribute to the effectiveness of the method.
> > - Insight into Design Choices. Provide deeper insights into the design choices made for the GCN architecture, particularly in relation to the V2X-Traj dataset. This includes discussing the rationale behind the choice of edge encoding, node feature extraction, and the inclusion of graph link prediction. How do these design choices specifically address the challenges posed by the cooperative motion forecasting task?

---

> > > ### Author Response · Authors · 2024-08-12
> > > **Official Comment by Authors**
> > >
> > > **1. Clarification.**\
> > > The proposed V2X-Graph is transformer-based rather than GCN-based. To address your concern, we will discuss our solution to the challenges of cooperative motion forecasting and compare it with existing graph-based methods below.
> > >
> > > **2. How V2X-Graph addresses challenges in cooperative motion forecasting.**\
> > > To leverage cooperative data effectively, two critical issues must be addressed: 1) Observations of the agents from different views may different due to various sensor perspectives and configurations; 2) In the cooperative scenario, there are multi-view observations of multi-agents, and the redundant data need to be leveraged interpretably.
> > >
> > > To address challenges and achieve better performance in cooperative motion forecasting, we focus primarily on motion and interaction features.
> > > **For motion features, we formulate them as differential information from each view, and aggregate them with spatial-temporal correlations, namely ST edge encoding, which can mitigate the deviation of direct single-frame individual fusion.**
> > > For interaction features, we consider features from all views and aggregate them with relative spatial information, namely RS edge encoding.
> > >
> > > To aggregate rebundant data interpretably, we introduce graph link prediction in our task, served as a guidance to fuse motion and interaction feature.
> > > Specifically, **for each agent, to leverage data from other views interpretably, we fuse the motion feature depicting the same agent, and we fuse the interaction feature of other agents, respectively, where the graph link prediction helps.**
> > >
> > > **3. Comparison with existing graph-based methods.**\
> > > There are two major differences with V2X-Graph.
> > >
> > > Existing graph-based methods mainly extract the motion feature for each agent as a node and formulate interactions with surroundings using full edge connections.
> > > In cooperative motion forecasting, however, there are redundant trajectories depicting agents from other views.
> > >
> > > **Existing methods lack a cross-view motion feature fusion design**.
> > > For trajectories depicting the same agent in other views, it is not reasonable to fuse interaction features.
> > > Instead, we design cross-view motion fusion to aggregate observations of the same agent from different views, which has been shown to be effective in ablation studies (Table 4 on page 8).
> > >
> > > **Existing methods lack an interpretable feature fusion design**.
> > > As shown in ablation studies (Table 5 on page 9), there is no obvious improvement between cooperation and ego-view settings when using full edge connections.
> > > This phenomenon is similar to that observed with existing graph-based methods.
> > > Instead of using full edge connections, V2X-Graph achieves better accuracy and computational efficiency improvement through the interpretable feature fusion of motion and interaction features.
> > >
> > > **4. Relationship between V2X-Graph and the V2X-Traj dataset.**\
> > > The proposed V2X-Graph does not have a tailored design for V2X-Traj.
> > > Actually, we design and evaluated the method on V2X-Seq at first.
> > > To further assess performance in a wider range of cooperative scenarios, we construct the V2X-Traj dataset.
> > > Experimental results on V2X-Traj demonstrate the effectiveness of the method in V2V and further V2X cooperation with three views.

---

### Author Rebuttal · Authors · 2024-08-07

## General Rebuttal for Commen Concerns
We will respond to the common concerns raised by the reviewers here.

**1. Our V2X-Graph is a pioneering work exploring trajectory-based feature fusion for cooperative motion forecasting.**

Most of the existing works in cooperative autonomous driving community pay more attention on cooperative perception, especially the single-frame cross-view feature fusion methods.
However, cooperative forecasting has not been well explored, which is a direct safety-related downstream task.
Existing perception-based methods obtain the agent state at each frame individually, which can not model the temporal motion and interaction feature for forecasting tasks, leading to performance degradation in downstream tasks.

The proposed V2X-Graph exploring forecasting-oriented trajectory feature fusion, which considers cooperative data holistically and integrates temporal motion and interaction features interpretably and in an end-to-end manner.


**2.1 Our V2X-Graph is robust to communication latency and data synchronization.**

We conduct robustness experiments on V2X-Seq dataset, taking both data synchronization problem and communication latency into consideration. Specifically, we simulate latency by dropping the latest one or two frames of infra-view data during transmission. And we address the latency issue with a simple interpolation to obtain synchronized trajectory data. Experiment results in the following table show that there is little performance degradation to communication latency.

| Latency (ms) | minADE | minFDE | MR |
| --- | --- | --- | --- |
| 0 | 1.0458 | 1.7855 | 0.2527 |
| 100 | 1.0468 | 1.7879 | 0.2562 |
| 200 | 1.0779 | 1.8385 | 0.2688 |
Table 1. Robustness experiment results on latency.

**2.2 Our V2X-Graph is robust to data loss and sensor failures.**

Data loss and sensor failures are common practical challenges. The performance advantage on real-world datasets demonstrates the robustness of our method. We furhter evaluate V2X-Graph under different data loss ratios on V2X-Seq dataset. Specifically, we randomly drop perception results data in each frame from the infra-view in transmission with various dropping ratios.
As shown in the table, the forecasting performance decreases as the ratio increases, but it is worth mentioning that our method outperforms the compared methods (table 1 on page 8) even under extreme conditions with a 50% loss rate.

| Loss Ratio (%) | minADE | minFDE | MR |
| --- | --- | --- | --- |
| 0 | 1.05 | 1.79 | 0.25 |
| 10 | 1.05 | 1.81 | 0.26 |
| 30 | 1.08 | 1.85 | 0.27 |
| 50 | 1.10 | 1.88 | 0.28 |
Table 2. Robustness experiment results on data loss.

**2.3 Our V2X-Graph is more efficient than other methods according to inference latency.**

Firstly, the parameter size of V2X-Graph is comparable with other motion forecasting methods, such as DenseTNT and HiVT.

| Dataset | Method | | |
| --- | --- | --- | --- |
| V2X-Seq | DenseTNT | HiVT | V2X-Graph |
| Param. (M) | 1.0 | 2.6 | 5.0 |
| V2X-Traj | DenseTNT | HDGT | V2X-Graph |
| Param. (M) | 1.0 | 12.1 | 4.9 |
Table 3. Parameter size comparison.

Then we conduct the inference experiment on single NVIDIA GTX 4090 and compare the inference cost. As for the experiment results shown, the proposed V2X-Graph is even faster than the compared vanilla motion forecasting methods, benefiting from the synchronous temporal state modeling and integration.

| Dataset | Method |  |  |
| --- | --- | --- | --- |
| V2X-Seq | PP-VIC + DenseTNT | PP-VIC + HiVT | V2X-Graph |
| Latency (ms) | 161.45 + 371.38 | 161.45 + 53.30 | **51.50** |
| V2X-Traj | DenseTNT | HDGT | V2X-Graph |
| Latency (ms) | 168.88 | 1260.70 | **52.69** |
Table 4. Inference cost comparison.

**3. We provide more additional details of the proposed V2X-Traj dataset.**

The location and time distribution of the proposed V2X-Traj dataset are presented in the PDF.

To avoid potential bias in the behavior patterns of autonomous vehicles, which leads to a distribution shift in the dataset, we deploy roadside sensors at 28 challenging and representative urban intersections and allow multiple autonomous vehicles to drive independently in these areas (Figure 1 in PDF).
A scenario is collected when the perception ranges of two vehicles and a roadside device overlap.

To ensure the diversity of traffic and weather conditions, the data is collected throughout the whole year (Figure 2 in PDF).

---

### Decision · Program_Chairs · 2024-09-25

**Decision:**

Accept (poster)

**Comment:**

The authors present V2X-Graph, a framework for interpretable and end-to-end trajectory feature fusion for cooperative motion forecasting. An advantage of this approach compared to the state-of-the-art in this area is that it focuses on the historical motion and interaction features rather than single-frame perception. Multiple reviewers raised the concern that it is not clear how the proposed method differs from existing GCN methods. They also acknowledged the response of the authors on this point. Additionally, the point was raised whether the V2X-Traj dataset might have inherent biases benefiting the proposed method. The authors have better explained the rationale behind the dataset (that represents a challenging scenario) and have given additional details on the dataset’s properties. Furthermore, the authors have addressed comments on the inference cost and the robustness under non-ideal communication conditions in their rebuttal.

All in all, the article represents a valuable contribution to the field and of sufficient level for presentation at NeurIPS. As there were quite some shared comments between the reviewers, which were addressed in the rebuttal, I would strongly advise the authors to modify the article according to the comments. This said, there were some expressed desires for more comparison and / or inclusion of more agents in the evalution, but since the current study is already considered very solid and quite extensive, I think these recommendations should be more taken onboard for future work.